# Stress Responsiveness and Emotional Eating Depend on Youngsters’ Chronic Stress Level and Overweight

**DOI:** 10.3390/nu13103654

**Published:** 2021-10-19

**Authors:** Kathleen Wijnant, Joanna Klosowska, Caroline Braet, Sandra Verbeken, Stefaan De Henauw, Lynn Vanhaecke, Nathalie Michels

**Affiliations:** 1Department of Public Health and Primary Care, Campus UZ-Ghent, Ghent University, Corneel Heymanslaan 10, 9000 Ghent, Belgium; Joanna.Klosowska@UGent.be (J.K.); Stefaan.DeHenauw@UGent.be (S.D.H.); Nathalie.Michels@UGent.be (N.M.); 2Laboratory of Chemical Analysis, Ghent University, Campus Merelbeke, Salisburylaan 133, 9820 Merelbeke, Belgium; Lynn.Vanhaecke@UGent.be; 3Department of Developmental, Personality and Social Psychology, Ghent University, Henri Dunantlaan 2, 9000 Ghent, Belgium; Caroline.braet@ugent.be (C.B.); Sandra.Verbeken@UGent.be (S.V.); 4School of Biological Sciences, Institute for Global Food Security, Queen’s University, 19 Chlorine Gardens, Belfast BT9 5DL, UK

**Keywords:** psychosocial stress, emotional eating, overweight, obesity, stress responsiveness, stress reactivity, cortisol reactivity, adolescents

## Abstract

The persistent coexistence of stress and paediatric obesity involves interrelated psychophysiological mechanisms, which are believed to function as a vicious circle. Here, a key mechanistic role is assumed for stress responsiveness and eating behaviour. After a stress induction by the Trier Social Stress Test in youngsters (*n* = 137, 50.4% boys, 6–18 years), specifically those high in chronic stress level and overweight (partial η^2^ = 0.03–0.07) exhibited increased stress vulnerability (stronger relative salivary cortisol reactivity and weaker happiness recovery) and higher fat/sweet snack intake, compared to the normal-weight and low-stress reference group. Stress responsiveness seems to stimulate unhealthy and emotional eating, i.e., strong cortisol reactivity was linked to higher fat/sweet snack intake (β = 0.22) and weak autonomic system recovery was linked to high total and fat/sweet snack intake (β = 0.2–0.3). Additionally, stress responsiveness acted as a moderator. As a result, stress responsiveness and emotional eating might be targets to prevent stress-induced overweight.

## 1. Introduction

In our current society, obesity and stress present major public health challenges [1] with 20% of adolescents globally suffering from mental health problems [2] and over 18% suffering from overweight and obesity [3]. Both stress and obesity are associated with psychological and physiological comorbidities [1,4,5,6]. Their shared involvement in multiple noncommunicable diseases is not surprising, as their interrelated nature via many pathways has been reported [1]. While clinical manifestations of these diseases mostly emerge later in life, strong evidence suggests that the effects of chronic stress and overweight trigger the disease pathways as early as during childhood and adolescence with long-lasting and frequently irreversible consequences [7].

Distress is a threatened emotional condition that initiates physiological, cognitive, and behavioural adaptive responses to maintain homeostasis [1]. Chronic stress, characterised by repeated threat and followed by inadequate or excessive responses, can lead to consistently altered arousal levels, called allostatic load [6,7,8]. As one of the psychosomatic changes, chronic stress can lead to weight gain via multiple pathways, some exposing a direct and others an indirect effect [4]. Of utmost importance is the immediate activation of neuroendocrine systems, with the two main axes being the autonomic nervous system (ANS) and the hypothalamic–pituitary–adrenal (HPA) axis. In a direct way, chronic activation of these systems and the subsequent hypersecretion of stress hormones (especially cortisol) have been shown to result in increased fat storage [6,7,9]. Indirectly, stress influences biological and behavioural pathways, which may lead to overweight via increased energy intake. Indeed, stress can undermine cognitive and behavioural processes [1,4] involved in altered eating behaviour, referred to as emotional eating [10]. Eating as a coping strategy in response to moderate levels of (chronic) stress has been observed in both laboratory and free-living conditions in all age groups [11,12,13]. This “emotional eating” behaviour eventually results in either overeating or enhanced unhealthy eating, i.e., eating foods high in fat and/or sugar in response to stress [4,8,14,15,16,17]. In addition, cortisol indirectly influences appetite regulation and reward processes. Cortisol is able to reduce the brain’s sensitivity to leptin, thereby inhibiting leptin’s satiety mechanism [1,6]. The dopaminergic or reward system on the other hand, stimulates the secretion of hunger-inducing ghrelin and neuropeptide Y [1,6,9] and the consumption of highly palatable foods in absence of hunger, a mechanism referred to as food wanting [18]. To sum up, all these mechanisms are linked to overeating and have direct or indirect deleterious effects by increasing adiposity.

In return, overweight and obesity are described as stressful due to prejudice and discriminative feelings towards increased body sizes. The latter often results in depression, anxiety, and eating disorders [1,4,6,9], resulting in increased levels of cortisol [1]. This mechanism where stress leads to increased adiposity and finally overweight/obesity, which in turn leads to more stress, is often referred to as the vicious circle of stress and obesity [1,9].

The existence of these interrelated mechanisms suggests that childhood chronic stress and obesity are intertwined, whereby changes in stress responsiveness (i.e., altered stress physiology and reports in response to a stressor) and eating behaviour (i.e., emotional eating) coexist. This interplay might be one of the reasons why obesity treatment is rarely successful [1]. Interestingly, Roemmich (2011) observed that not all children with obesity increase their energy intake when stressed, but that individual differences in psychological behaviour can act as moderators [17]. Hence, stress responsiveness might be a vulnerability factor that determines whether chronic stress will lead to weight gain or not.

As research on this topic is limited, especially in youngsters, we wanted to broaden the knowledge on the interrelationship of chronic stress and obesity by providing an answer to the omnipresent question “Why do certain adolescents with high stress gain weight and others not?” and, more specifically, “Do high stress reactivity and related emotional eating distinguish stressed overweight youngsters from stressed youngsters without overweight?”. Therefore, we aimed to study the differences in stress responsiveness (during an acute laboratory stressor) and emotional eating (during a snack exposure in the lab after the stressor) among youngsters varying in weight status and chronic stress levels. First, we hypothesised that mainly youngsters with both elevated chronic stress levels and overweight would express a stronger stress reactivity, weaker stress recovery, and more emotional eating in response to an acute lab stressor. After all, these three factors can stimulate adiposity and, thus, might help in identifying a subset of stressed individuals with high overweight risk. Herein, different stress responsiveness markers were distinguished to elucidate mechanistic differences. Second, we expected that emotional eating in the lab would be explained by high stress reactivity (as stress physiology and cognitive stress processes stimulate the urge to eat) and by trait emotional eating (to verify momentary lab reaction, i.e., state with personality trait). In an explorative analysis, we expected that stress reactivity would be a moderator in lab-based emotional eating differences between overweight/stress groups, i.e., that group differences in emotional eating would become more apparent in cases of high stress reactivity since high stress reports or hormones stimulate emotional eating.

## 2. Materials and Methods

### 2.1. Participant Sample

A total of 141 Belgian children and adolescents (50.4% boys, 6–18 years) were recruited from the Obesity Prevention through Emotion Regulation in Adolescents (OPERA) study (*n* = 123) [19] and from the paediatric obesity department of the Jan Palfijn hospital in Ghent, Belgium (*n* = 18). The additional recruitment of the second group was to provide an optimal context for studying differences in adiposity by oversampling youngsters with obesity. Recruitment and sample collection took place from March until August 2018.

Inclusion criteria included no severe underweight (BMI was calculated and a BMI *z*-score to adjust for sex and age was calculated using to the Flemish reference data) [20]. Cut-offs for severe underweight were based on the International Obesity Task Force (IOTF) criterion [21], e.g., BMI *z*-score < −1.014 for boys and BMI *z*-score < −0.975 for girls, *n* = 1), no endocrine diseases (two youngsters with diabetes and one with hypertension were excluded), and no concomitant administration of glucocorticoid medicines (*n* = 0), resulting in 137 participants. The study was approved by the Ethics Committee of the Ghent University Hospital (EC2017/0527) and in accordance with the Helsinki Declaration. Written informed consent was obtained from parents and adolescents (≥12 years). Children below the age of 12 provided verbal assent to participate.

### 2.2. Experimental Design

The study was divided into two phases and was executed from March until August 2018.

The first phase involved an online questionnaire which each participant and one guardian had to complete. The guardian questionnaire included demographic questions, the child’s age, the child’s medical history, and the parents’ educational level. The participants were questioned online about their sex, pubertal stage (Tanner scale [22]), experienced and perceived stress, depressive symptoms as an indicator of mental health, and eating behaviour.

During the second phase, adolescents were invited to the Ghent University Hospital Campus Research Centre. Appointments were only made on school days, outside school hours (Monday, Tuesday, or Thursday starting between 4:30 and 5:50 p.m., Wednesday starting between 2 and 2:30 p.m. and between 4:30 and 5:50 p.m.). We assumed that, at these timepoints, feelings of tiredness and hunger would be similar as all participants had to go to school and were invited outside eating hours. In order to not affect snack intake during the food lab, participants were asked to abstain from eating or drinking (except water) at least 2 h before their appointment. For the purpose of standardising the intervention procedure, an instruction video was developed and shown to all participants during the intervention—starting at T1 and ending at T6 (Figure 1). This video followed a strict time schedule, providing the participants with stepwise instructions for each new phase in the process (including all relaxation moments, the completion of the online questionnaires (Q1 to Q5), the sample collections (S1 to S6), the TSST-C, and the snack buffet). Additionally, a trained researcher was available to help the participants. Before the actual start of the experimental protocol, participants received a preload (a plate with four small ham or cheese sandwiches) and were allowed to eat until satisfied. In the meantime, the heart rate variability (HRV) measuring device was installed on the adolescent’s upper body for continuous measurement. The participants had to stay seated, with both feet on the floor and knees at a 90° angle. The laboratory experimental protocol was initiated with the start of the video and the HRV recording, and it was divided into different data collection phases (Figure 1). To ensure low levels of stress at the start, two 7 min relaxation moments were foreseen: one without cognitive load (a) and one with a validated relaxation movie on nature in Alaska (b) [23]. The main parts of the experiment were TSST-C [24], a snack buffet food lab, saliva collections (S1–S6) for cortisol (sCortisol) and alpha-amylase (sAA), and questionnaires (Q1–Q5). Depending on the experimental phase, different questions were asked on hunger, taste, food wanting, stress, and mood. At the end of the study (after the experimental protocol), the weight and height of the participants were measured.

### 2.3. Stress Manipulation

At present, the Trier Social Stress Test (TSST) is the gold standard for human experimental stress induction. To sufficiently stimulate the two different stress axes, it consists of a speaking and an arithmetic task [25,26]. To study youngsters’ stress reactivity and emotional eating behaviour in response to an acute stressor, the TSST adjusted for children (TSST-C) was used [8,24,25]. For the first task, participants were asked to finish a “scary” story with an open ending instead of applying for a job. The second, arithmetic task was adjusted in level of difficulty on the basis of age. Both tasks were held in front of a camera, with a neutral-looking investigator present.

### 2.4. Response to the Stressor: Overall Responsiveness, Stress Reactivity, and Stress Recovery

Response to the stressor was measured via changes in sCortisol, sAA, HRV, and state measures of stress and emotional experience, as different biomarkers and reports might uncover different mechanisms. (1) To reflect overall *stress responsiveness*, the area under the curve with respect to increase (AUCi) was calculated on the basis of published formulas, hereby emphasising the changes in physiological stress responses over time and correcting for individual differences at baseline (sCortisol2–6, sAA2–5, HRV2–4, and Q2–4) [27]. (2) *Relative stress reactivity* was measured from start stressor (sCortisol2, sAA2, HRV2, and Q2) to peak of stress induction (sCortisol4, sAA3, HRV3, and Q3), as a percentage compared to start value. (3) *Relative stress recovery*, given as percentage, was calculated by subtracting the stress peak from the expected recovery (sCortisol6, sAA5, HRV4, and Q4) and compared to the peak value. Calculations started from the second and not from the first baseline value (sCortisol1, sAA1, HRV1, and Q1), since participants still had to get used to the new environment and the investigator (Appendix A shows indeed a decreased stress report and salivary cortisol between TSST–28 and TSST start).

#### 2.4.1. Salivary Cortisol and Alpha-Amylase

sCortisol accurately represents the active component of circulating cortisol in the blood [28], while sAA is used as biomarker reflecting the ANS (sympathetic) mediated stress response. Cortisol shows a 10 to 15 min delay in response to stress and a slow return to baseline (i.e., 50 min after baseline) [8,26,29]. To compensate for this delay, participants were asked to collect saliva at six different timepoints throughout the study (see Figure 1): baseline (S1), stressor onset (S2), stressor end (S3), and three post-stress measures (S4–S6). The time point TSST + 27 min (S4) was considered the cortisol peak, while TSST + 55 min (S6) was considered the endpoint. Saliva samples were collected using Salivette synthetic swabs (Salivette, Sarstedt, Germany), immediately placed on dry ice and stored at −20 °C. Cortisol was assessed via an enzyme-linked immunosorbent assay (ELISA) with an inter-assay CV of 7.2%, intra-assay CV of 6% and range of 50–3200 pg/mL (linear interval 100–1600 pg/mL) (Arbor Assays^®^) [30]. The values that fell outside the standard curve were reported as system missing, but the values outside the linear interval were kept by maintaining enough data points to observe the expected peak in cortisol concentration across the different intervals. The activity of sAA was measured using the Amylase Activity Assay Kit from Sigma-Aldrich [31], with a standard curve range 0–20 nmol/well nitrophenol and an inter-assay CV of 5.4%.

#### 2.4.2. Heart Rate Variability

HRV represents the fluctuation in time intervals between consecutive heartbeats, helping humans to adapt to environmental and psychological challenges. Loss of this heart rate variability can predict mental and physical morbidities, due to an increased sympathetic output and decreased parasympathetic activity [32]. Elevated levels of vagus nerve-mediated resting HRV are associated with better emotion regulation [33]. In our study, HRV was measured using the compact 90° eMotion Faros (Mega Electronics Ltd. Finland) attached to the chest area via two electrodes (one at the middle of the right clavicle and the other at the left bottom side of the rib cage). During the measurement, participants were seated in a quiet environment and asked not to move. Kubios HRV Premium 3.3.1 was used for data processing after an automatic artefact correction to remove ectopic beats [34]. The root mean square of successive differences between normal heartbeats (RMSSD) was selected, primarily reflecting the parasympathetic activity and, thus, relaxed status [35]. To reflect sympathetic nervous system activity, the square root of Baevsky’s stress index (stress index) was used; higher values correlate with more stress [36]. Since stress reactivity measures were based on change from the baseline HRV, most inter-subject variation, e.g., in physical fitness, was already corrected for. As can be seen in Figure 1, six HRV segments were obtained: two baseline values, one during TSST-C for stress reactivity, and three for stress recovery. As chewing stimulates the ANS, the HRV measures taken after the snack buffet food lab were not considered, and HRV4 was, thus, used as the full-recovery point.

#### 2.4.3. Mood and Self-Reported Stress

We intended to measure stress-related mood changes before and after the stress induction as a reflection of subjective stress reactivity and recovery. Since the questioning was repeated four times and the stress measure was repeated five times during the overall experiment, a short version of the Positive and Negative Affect Schedule for Children (PANAS-C) [37] was chosen. The PANAS-C included four stress-related items: three negative affect items (sad, afraid, and mad) and one positive affect item (happy). These items were scored by the participants on a VAS-scale ranging from 1 “not at all” to 100 “very much”. The scores were recoded into one negative and one positive score. One additional item “I feel stressed” was scored accordingly.

### 2.5. Snack Liking/Wanting Score and Snack Buffet Intake during the Food Lab

#### 2.5.1. Snack Categories

To investigate the effect of an acute stressor on emotional eating and food choice, a food laboratory was included in the study. The setup was based on Oliver et al. [38] and a well-validated food preference tool [39] using four snack categories: high-fat savoury (HFSA), low-fat savoury (LFSA), high-fat sweet (HFSW), and low-fat sweet (LFSW). For each of the four categories, three snacks were chosen from the experimental research image database “Food-Pics” [18] on the basis of recognisability and familiarity scores above 90%.

#### 2.5.2. Liking and Wanting Score

An important differentiation was made between food liking (hedonic value), which is a process incorporating taste and involves sensory properties, as well as an individual’s physiological state and associative history, and food wanting (motivation), which is an implicit and objective drive process towards a demand for the targeted food item [39,40]. The pictures of all 12 snacks were randomly shown before and after the TSST-C for scoring (see Appendix A) to answer the questions for liking “In general, how much do you like this food?” and for wanting “How much would you like to eat this food now?”. To get an idea of overall wanting and liking, a sum score over all 12 snacks was calculated. Liking was tested as a confounder, while wanting increase after TSST-C was tested as a proxy for emotional eating. As the participants were told the experiment’s goal was to test changes in taste in response to stress, they were asked to taste each snack during question round Q3 (see Figure 1) after the TSST-C stress induction.

#### 2.5.3. Snack Buffet Intake

The total snack intake (in kcal and g/kcal) and snack intake per category (in kcal) were investigated after the participants were left alone for 10 min. Participants were told that they could eat the remaining snacks, while, in the meantime, the investigator would check the collected data before ending the study. On the table, 50 g of each snack was available in identical aluminium trays, allowing a total of 2494.5 kcal intake. To correct for being hungry, an additional question “Are you hungry?” rated on a VAS scale (1–100) was asked at study start and after the TSST-C stressor.

### 2.6. Weight and Chronic Stress Level as Predictor

Participants were divided into four BMI/stress groups according to IOTF score (normal weight versus overweight and obesity) and chronic stress sum score (based on median split): low/low (NWLS); low/high (NWHS); high/low (OWLS); high/high (OWHS).

#### 2.6.1. Body Composition Measures

Weight measures (up to 0.1 kg) were taken by the investigator using a bioelectrical impedance instrument (Tanita, The Netherlands), and length measures (up to 0.1 cm) were taken using a portable stadiometer (Seca, Germany). BMI was calculated, after which a BMI *z*-score to adjust for sex and age was determined using the Flemish reference data [20]. Weight cut-offs were based on the IOTF criterion [21], in which overweight is defined by *z*-scores between 1.310 and 2.288 for boys and between 1.211 and 2.192 for girls, along with obesity as a *z*-score ≥2.288 for boys and a *z*-score ≥2.192 for girls. The term overweight was further used to cover both phenomena.

#### 2.6.2. Chronic Stress Level

Chronic stress levels were assessed by three parameters reflecting both subjective and objective stress measures: the Perceived Stress Scale (PSS), the Children’s Depression Inventory (CDI-2), and hair cortisol. A summary chronic stress variable was calculated by adding up the three *z*-score-transformed variables to give them equal weight. High and low chronic stressed groups were based on the median split of the total sum score, as no clinical cut-offs are currently available. Factor analysis supported this one-factor structure, which explained 58% of all variance.

Perceived stress was reported by the participants through the 10-item Perceived Stress Scale (PSS) and reflected a period up to 1 month prior to questioning [41]. After reverse-coding certain questions (all scored 0–4), a sum score (α = 0.74) was calculated with higher scores, reflecting higher stress levels [41,42].

Depressive symptoms were reported by the participant via the Children’s Depression Inventory (CDI-2) [43,44] containing 28 questions with three options (values 0, 1, and 2). After reverse-coding certain questions, a sum score was calculated (α = 0.85). A higher sum score reflects more depressive symptoms and can be seen as an indicator of a mental health burden.

Hair cortisol was measured by taking hair samples from the vertex posterior scalp area with a required minimum length of 3 cm, as this length reflects the stress endured in the last 3 months (hair grows at a rate of approximately 1 cm/month). Subjects with hair shorter than 3 cm were excluded (*n* = 9) from this part of data collection. Liquid chromatography–tandem mass spectrometry was used [45] for quantitative cortisol analysis in hair with a limit of quantification (LOQ) equal to 1.57 pg/mg; samples measured at LOQ were assigned the value of the LOQ during statistical analysis. Higher hair cortisol levels indicate higher chronic stress levels, but no well-accepted cut-offs exist yet [46]. The use of hair cortisol for chronic stress measures is a completely different concept than the salivary cortisol reactivity measure for acute stress monitoring [46,47].

### 2.7. Trait Emotional Eating

Trait emotional eating might influence participants’ eating behaviour, especially during a food laboratory following a stressful event. Hence, participants were asked to complete the Dutch Eating Behaviour Questionnaire (DEBQ) [48]. The DEBQ is a 33-item measure of eating behaviours rated on a five-point Likert scale, with higher scores indicating more behavioural problems. In the present study, the emotional eating subscale (13 items) was utilised (α = 0.96).

### 2.8. Statistical Analyses

Analyses were performed in SPSS (IBM SPSS Statistics 24) with two-sided *p* < 0.05 as the significance level. The parameters hair cortisol, stress, negative emotions, sCortisol, and RMSSD were not normally distributed and were, therefore, log-transformed. Resulting estimated marginal means were back-transformed into original units for interpretation. Descriptive statistics included Spearman’s correlation to test interdependencies, one-way ANOVA to test group differences for continuous normally distributed variables (mean ± SD) and Kruskal–Wallis for continuous non-normally distributed variables (median (p25; p75)), and chi-square for categorical variables (*n*; %). An a priori sample size calculation indicated *n* = 88 for a small effect (partial η^2^ = 0.02) in time × group repeated-measures ANOVA, *n* = 159 for a small effect (f^2^ = 0.05) in linear regression, and *n* = 125 for a medium effect (partial η^2^ = 0.06) in interaction analysis.

First, repeated-measures ANOVAs were run for all the outcome measures (sCortisol, sAA, stress report, negative emotions, happy, stress index, and RMSSD) using time as within-subject factor, BMI/stress group as a between-subject factor, and their interaction time × BMI/stress. Adjustments were made for sphericity assumption violation. Estimated marginal means were used to create figures.

Second, linear regression analyses were run to test the four BMI/stress groups, the continuous BMI and chronic stress variables (single model, combined model, and with BMI × chronic stress as an interaction) as predictors of stress reactivity/recovery (based on calculated relative change and AUCi), emotional eating (state and trait), and food wanting. Multicollinearity (variance inflation factor >5) was absent in all models. All ANOVAs and regressions were adjusted for following factors: age, sex, and highest education level of parents. Analyses with snack intake and food wanting as outcome were additionally adjusted for hunger and food liking, both at study start. Linear regressions with stress recovery as a predictor and snack buffet response as an outcome were adjusted for the respective stress reactivity when the stress reactivity had a significant association with snack intake. Tukey’s test was used to adjust for multiple comparisons in post hoc analyses.

Moderation by stress responsiveness was tested with the PROCESS software [49] by testing the significance of the interaction “group × stress” term (after centring included variables). In case of a significant interaction (*p* ≤ 0.05), the relationship between stress-related independent variables and outcomes was tested at three levels of the particular moderator variable: at the 16th, the 50th and 84th percentiles. At each level, significant group differences were indicated compared to the reference group NWLS.

## 3. Results

### 3.1. Descriptive Data and Intervention Check

Descriptive data can be consulted in Table 1 and Appendix A. Overweight prevalence was 29%, of which 12 participants had obesity. For 21.2%, mild depressive symptoms (CDI > 13) were reported [44], and 70.1% had moderate stress levels (PSS > 13) [50]. The four BMI/stress groups did not differ in sex, Tanner stage, parental education, hunger at study start, total wanting, trait emotional eating, and most baseline stress parameters. Some significant group differences were found; OWLS had a lower age than NWHS, NWLS had an overall higher liking score than OWLS, and OWHS had lower sCortisol than all other groups.

Using repeated-measures ANOVA, significant changes over time (time effect *p* < 0.05) were seen during the laboratory intervention for all measures except happy report (Appendix A). Herein, post hoc tests with estimated marginal means showed significant changes by the TSST-C in the total sample, except for the measures happy, negative emotions (despite a significant recovery), and stress index. However, the reactivity values in Table 2 suggest that certain individuals still experienced reactivity in these three parameters. 

From baseline to peak, the stress report increased by 91% (*p* < 0.001), sCortisol increased by 154% (*p* < 0.001), sAA increased by 5.8% (*p* = 0.016), and RMSSD decreased by 12% (*p* ≤ 0.001) (see Appendix A). The TSST-C also induced increased food wanting (from 386 to 494 on a scale of 12–1200) and perceived hunger (from 13 to 25 on a scale of 1–100) (overall time effect *p* ≤ 0.001). The participants scored “liking of the presented snacks” 744 on a scale of 12–1200; therefore, the food laboratory offered enough attractive snacks. The average consumed intake was 355 kcal and ranged from 50 to 1276 kcal per person.

Intercorrelations between parameters of interest can be found in Appendix A.

### 3.2. Stress Reactivity/Recovery in Relation to Chronic Stress and Weight Status

Results on BMI/stress group differences in both stress reactivity and stress recovery (from stressor start until end of recovery) can be found in Appendix A (based on repeated-measures ANOVA) and in Table 2 (based on linear regression). An overall significant group difference was found for sCortisol, as reflected in a significant time × group interaction (*p* = 0.043, partialη^2^ = 0.068, medium effect size) in the repeated-measures ANOVA. Indeed, sCortisol’s relative reactivity (β = 0.152) was significantly higher in the OWHS group than in the NWLS and NWHS groups (partial η^2^ = 0.053 and 0.055). In addition, happiness recovery (β = 0.195) was weaker or non-existent (with even a further happiness decrease during recovery) in the OWHS compared to the NWLS group (partial η^2^ = 0.049). Continuous BMI *z*-score and chronic stress score as single predictors, as a combined model, or as an interaction could not significantly predict stress reactivity/recovery (Appendix A).

### 3.3. Food Lab Responses in Relation to Chronic Stress and Weight Status

Results of group differences in food lab response can be consulted in Table 2. A significantly higher kcal intake of HFSW snacks (β = 0.200, *p* = 0.037) was seen in the OWHS group in comparison to the NWLS (partial η^2^ = 0.060, medium effect size) and NWHS groups (partial η^2^ = 0.032).

Next, when considering continuous chronic stress and BMI individually or together as a predictor (Appendix A), only a significant increase in HFSW intake (kcal) was confirmed for higher BMI *z*-scores, and chronic stress was positively associated with trait emotional eating.

### 3.4. Stress Responsiveness as Moderator

Stress reactivity was tested as a moderator by testing the interaction between BMI/stress groups and stress reactivity in the prediction of emotional eating, especially in cases of high stress reactivity (Appendix A). Figure 2 shows only the significant moderating effect of stress reactivity (*R^2^* change ranges 4–9% = small-to-medium effect size). First, an overall significant interaction was seen by stress reactivity with LFSW as an outcome; OWLS had a higher LFSW caloric intake than NWLS in case of a high stress increase. Second, happiness reactivity was also a significant moderator toward total snack buffet intake and the caloric intake of HFSW and LFSW snacks. Compared to the reference NWLS, OWHS showed a small positive association between happiness reactivity and snack buffet caloric intake (total, HFSW, or LFSW).

Considering the hypotheses first and foremost focused on the effect of acute stress and, therefore, stress reactivity, the possible moderation effects of stress recovery and AUCi on the BMI/stress differences in snack buffet intake are described in Appendix A.

### 3.5. Interrelations between Stress Reactivity/Recovery and Food Lab Responses

A positive association was seen between the sCortisol reactivity and caloric intake of HFSW snacks (Appendix A). Concerning the ANS, the stress index recovery was positively (i.e., less recovered) associated with the total caloric intake of snacks, as well as with the HFSW and LFSA caloric intake. A logically opposite effect was seen for RMSSD; higher RMSSD recovery (i.e., better recovery) was associated with lower total caloric intake, as well as HFSW and LFSA caloric intake. Moreover, the increase in wanting (i.e., state emotional eating) was related to higher stress levels due to less recovery; a positive association was seen with stress-report relative recovery and a negative association with RMSSD recovery.

### 3.6. The Role of Trait Emotional Eating

Self-reported emotional eating was not different across the four BMI/stress groups (Table 1 and Table 2) and was not associated with BMI *z*-score, although a positive association was seen with chronic stress *z*-score (Appendix A). Contrary to expectations [51], self-reported emotional eating was not associated with the laboratory stress response or with TSST-C induced snack consumption (Appendix A for correlations, Appendix A for regressions).

## 4. Discussion

Via a stress task and experimental snack buffet, this mechanistic study investigated the difference in stress responsiveness and emotional eating among youngsters varying in weight and chronic stress levels. We demonstrated that (a) specifically youngsters with a combination of high stress and overweight experience higher stress reactivity (i.e., salivary cortisol reactivity) and more emotional eating after stress (i.e., only high sweet and fat snacks), (b) emotional eating in the lab can be explained by high stress reactivity, and (c) stress reactivity has a moderating effect on lab-based emotional eating differences between weight/stress groups. 

This research confirmed that a subgroup of youngsters presenting a combination of chronic stress and overweight experience an increased stress responsiveness and intake of highly palatable snacks following exposure to a laboratory stressor (TSST-C). Interestingly, the snack intake diversity observed in this subgroup was restricted to eating more high-fat and sweet snacks and not overall snacking or snacking of savoury or low-fat snacks. Multiple studies described such significant associations between chronic stress and greater preference for energy-dense foods, i.e., those high in sugar and fat [4,9,52,53]. This rather “unhealthy” eating behaviour further leads to weight gain [1]. When not treated appropriately, this emotional/stress eating behaviour might result in overweight. Not surprisingly, our data showed a positive association between BMI and high-fat sweet snack intake. Unexpected, however, was the lack of a positive association between chronic stress and palatable snack intake. This might imply that specific mechanisms are in place that may clarify why only a certain subgroup and not all stressed youngsters are overweight.

Indeed, we hypothesised that a key role might be assigned to stress responsiveness. Furthermore, we believed that a population of youngsters can be stratified on the basis of their stress responsiveness, i.e., that higher stress reactivity and less stress recovery might lead to more emotional eating and subsequent weight gain. Our data confirmed that, compared to the reference group, youngsters presenting a combination of chronic stress and overweight experienced a less beneficial stress responsiveness following exposure to a laboratory stressor. This higher stress reactivity was observed as a stronger relative increase in salivary cortisol, a marker of the HPA stress axis [6]. Importantly, chronic stress or overweight as single or combined continuous predictors were not significantly related to higher salivary cortisol reactivity or other tested stress responsiveness markers. Hence, it may be deduced that not all adolescents with chronic stress or overweight are highly sensitive to an acute stressor but only a specific subgroup with both features. This finding was in line with a recent study in adults (healthy controls versus obesity), in which an acute stressor increased the levels of cortisol only in a subgroup of individuals with obesity (i.e., those categorised as high cortisol reactors) [54]. Apart from (epi)genetic predisposition, one explanation might be that these high cortisol reactors are only visible in the overweight group suffering from chronic stress, suggesting they cannot cope with stress. Indeed, a review by Tomiyama described that high stress-induced cortisol was not present in all individuals with obesity, but only in those with high-weight stigma [1], which might be a proxy for stress. A second explanation for the observation of this subgroup might be that a combination of high stress reactivity and related emotional eating drives these stressed adolescents towards overweight. Indeed, high stress hormone levels due to high stress reactivity would render these individuals more prone to stress eating, leading to an increased energy intake and, thus, indirectly to weight gain [4]. Moreover, higher stress hormone levels may also directly boost fat storage [6]. Unique was the finding that, in our sample of youngsters, high salivary cortisol reactivity but not chronic stress was associated with increased high-fat and sweet snack intake, thereby uncovering one possible mechanism explaining the correlation between acute stress and altered eating behaviour. Indeed, cortisol secretion is described to stimulate appetite and increase body weight by inhibiting leptin (an important regulator of energy homeostasis, reward processing, neuroendocrine functioning, and metabolism), as well as stimulate the orexigenic action of ghrelin and neuropeptide Y [1,9]. This stress-induced eating or emotional eating generally leads to unhealthy eating behaviours or choices and not increased overall eating [4] as reflected in only higher intake of sweet and fat foods by the stressed overweight group. Recent studies in adults support the finding that high cortisol reactivity is a predictor of higher sweet snack consumption [53,54,55]. The current study highlights that, also in youngsters, cortisol reactivity is linked to high-fat and sweet food intake after a laboratory stressor, and that this might be one of the reasons why chronic stress only leads to overweight in a subgroup, i.e., in those with high stress reactivity and emotional eating. This central finding seems to imply that differential treatment strategies should be adapted to high-risk subgroups. This statement is in agreement with Evers (2018) who emphasised the need for specific subgroups (i.e., restraint eaters, eating disorders, etc.) when studying altered eating behaviour [56]. Applied to the current findings, although caution should be taken considering our laboratory study design, overweight treatment should integrate a stress coping mechanism for those with high chronic stress levels or stress reactivity. Currently, overweight treatment is mainly focused on general dietary advice and physical activity stimulation, while stress/emotion regulation can also play a causal role [23,56,57,58].

Less commonly studied stress markers were also included in this research to have a better understanding of the physiological pathways involved in stress responses. Apart from the HPA axis, the ANS responds rapidly to stressors, with the sympathetic ANS assisting arousal for basic bodily requirements and the parasympathetic ANS exerting predominantly restorative functions [59]. Although no significant responsivity difference was found for both HRV parameters (i.e., stress index and RMSSD) between the stress/weight groups, youngsters with a less effective ANS recovery (i.e., higher arousal levels after the stress peak and a slower return to baseline) increased their total and unhealthy snack intake. These associations demonstrate that, in addition to cortisol, changes in the ANS are related to increased palatable food intake. Monitoring HRV might be a valuable non-invasive method that does not require collection of human body fluids to observe changes in stress responsiveness after an acute stressor. As stress is not the only reaction individuals perceive after an acute stressor, we expect that behavioural changes were not solely the result of perceived stress and, therefore, other emotions (positive or negative) might be involved. Indeed, happiness and stress are described to be interdependent as they often accompany one another. Furthermore, altered feelings of happiness have been linked to increased food intake due to impaired cognitive control [56]. Our research showed that youngsters high in both chronic stress level and BMI had a weaker happiness recovery, which was recorded as stable or worsening feelings of happiness. Although, in our study, no direct associations were found between happiness responsiveness and snack intake, the results confirm the involvement of multiple emotions after an acute stressor.

In addition, we hypothesised a moderating role for stress responsiveness towards stress-induced eating when studying group differences (stress/BMI) as predictor. Indeed, our moderation analyses provided a first proof that higher reactivity (in self-reported happiness and stress) led to a higher intake of sweet/fat snacks or lower intake of the healthier low-fat savoury snacks in at-risk groups. This finding did not specifically uncover more risks for the high-stress overweight group as hypothesised, but the increased vulnerability to emotional eating by high stress responsiveness was also seen in those participants who presented only high stress or only overweight (without the combination). To the best of our knowledge, other studies investigating the moderating effect of stress responsiveness on eating behaviour are lacking. Overall, the current moderation results seem to highlight the relevance of targeting stress responsiveness (a vulnerability factor) in interventions to decrease health effects of chronic stress exposure in current society. Indeed, training adaptive stress-coping via emotion regulation to avoid emotional eating seems essential [23,57].

A set of study strengths and limitations involved in the research project is described in Appendix A.

## 5. Conclusions

This stress induction and snack laboratory study provides novel insights into stress–overweight mechanisms in youngsters via stress responsiveness and emotional eating. Only youngsters with both chronic stress and overweight experienced high stress responsiveness (strong cortisol reactivity and no happiness recovery and, thus, high stress-induction vulnerability) and emotional eating of energy-dense foods. Indeed, highly stress-responsive youngsters ate more palatable foods, a mechanism stimulating weight increase. Exploratory moderation analyses further suggested that high stress reactivity and poor stress recovery can sometimes lead to more and/or unhealthy food intake after stress in at-risk groups. These findings are particularly important towards overweight treatment; a specific overweight group of high stress-responders is in need of stress interventions on top of currently advised energy intake (via dietary advice) and expenditure (via physical activity) adaptations. Furthermore, training adaptive stress-coping via emotion regulation and avoiding emotional eating also seems essential to overweight prevention. Our novel insights should be confirmed in longitudinal studies (again using diverse stress measures), while emotion regulation intervention studies are warranted to corroborate the evidence. 

## Figures and Tables

**Figure 1 nutrients-13-03654-f001:**
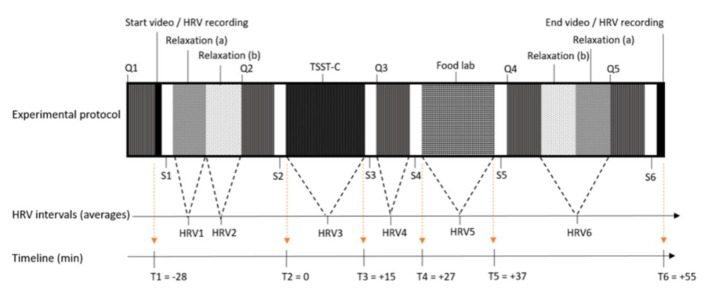
**Experimental design of the study.** Experimental design including the main parts and timeline: time in min relative to the Trier Social Stress Test for Children (TSST-C) (T1-T6), questionnaires (Q1-Q5), saliva collections (S1-S6), and heart rate variability measurement intervals (HRV1-HRV6).

**Figure 2 nutrients-13-03654-f002:**
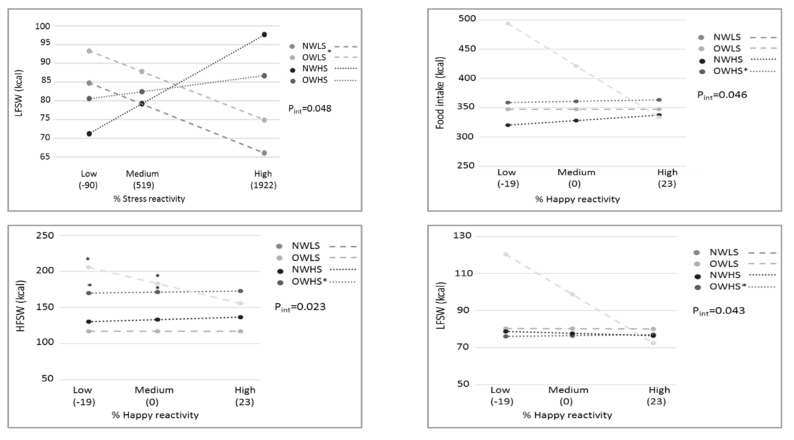
Moderating effects of stress reactivity measures on associations between BMI/stress groups and snack buffet intake after the stressor. * Significant group differences for the stress reactivity versus snack buffet intake relation or for snack buffet intake differences at specific stress reactivity levels, with NWLS as the reference group. P_int_ = *p*-value of the group × reactivity interaction; NWLS = normal weight low stress; OWLS = overweight low stress; NWHS = normal weight high stress; OWHS = overweight high stress; HF: high fat; LF: low fat; SW: sweet.

**Table 1 nutrients-13-03654-t001:** Descriptive data and BMI/stress group differences at baseline.

	Normal Weight and Low Stress *n* = 52	Overweight and Low Stress *n* = 16	Normal Weight and High Stress *n* = 46	Overweight and High Stress *n* = 23	Total Sample *n* = 137	*p*-Value Overall Group Difference
Body mass index *z*-score	−0.2 ± 0.6 ^a,b^	1.9 ± 0.7 ^a,c^	−0.0 ± 0.7 ^c,d^	1.7 ± 0.6 ^b,d^	0.4 ± 0.4	**<0.001**
Weight category (*n*; % overweight or obese)	52; 0% ^a,b^	16; 100% ^a,c^	46; 0% ^c,d^	23; 100% ^b,d^	137; 28.7%	**<0.001**
Perceived stress scale (0–40)	12.8 ± 3.5 ^a,b^	13.8 ± 4.9 ^c,d^	20.5 ± 4.0 ^a,c^	20.0 ± 4.0 ^b,d^	14.2 ± 2.0	**<0.001**
Depressive symptoms (0–56)	5.0 ± 3.4 ^a,b^	6.0 ± 4.4 ^c,d^	12.0 ± 6.2 ^a,c^	14.1 ± 5.2 ^b,d^	9.0 ± 6.2	**<0.001**
Hair cortisol (pg/mg)	2.1 (1.6; 2.7) ^a,b^	1.8 (1.6; 2.2) ^c,d^	3.1 (2.7; 5.8) ^a,c^	3.7 (2.5; 5.1) ^b,d^	2.6 (1.9; 3.7)	**<0.001**
**Background**						
Sex (*n*; %women)	20; 38.5%	8; 50%	28; 62.2%	13; 56.5%	69; 50.7%	0.121
Age	14.0 ± 1.6	13.2 ± 2.9 ^a^	14.8 ± 1.6 ^a^	13.9 ± 2.4	14.2 ± 2.0	**0.030**
Tanner stage (*n*; % prepubertal)	1; 1.9%	2; 12.5%	3; 6.7%	4; 17.4%	10; 7.4%	0.098
Parental education (*n*; % post-secondary)	30; 57.7%	7; 43.8%	25; 55.6%	10; 43.5%	72; 52.9%	0.580
Trait emotional eating (13–65)	26.5 ± 13.5	25.4 ± 10.6	30.6 ± 10.6	26.1 ± 10.5	27.7 ± 11.8	0.245
**Stress Parameters at Start**						
Stress report (1–100)	6 (1; 13)	3 (1; 10)	10 (1; 20)	10 (1; 28)	10 (1; 20)	0.124
Negative emotions report (3–300)	3 (3; 12)	3 (3; 7)	3 (3; 17)	8 (3; 24)	3 (3; 15)	0.271
Happy report (1–100)	69.9 ± 25.2	71.8 ± 20.0	63.3 ± 20.8	58.4 ± 27.3	66.1 ± 23.7	0.209
HRV RMSSD (ms)	51 (36; 77)	63 (42; 82)	53 (41; 69)	48 (34; 34)	51 (39; 76)	0.771
HRV stress index	8.3 ± 3.2	3.2 ± 3.6	8.1 ± 3.1	9.0 ± 2.9	8.3 ± 3.1	0.835
Salivary cortisol (pg/mL)	309 (142; 558) ^a^	553 (249; 734) ^b^	441 (160; 816) ^c^	116 (75; 263) ^a,b,c^	311 (115; 693)	**0.021**
Salivary alpha-amylase (nmol/min/mL)	46,265 ± 18,284	35,549 ± 20,108	44,776 ± 21,262	40,929 ± 17,851	43,903 ± 19,562	0.306
**Food Parameters at Start**						
Hunger (1–100)	19.4 ± 21.7	5.8 ± 7.4	19.3 ± 21.7	8.6 ± 13.2	16.4 ± 20.2	**0.038**
Liking (12–1200)	791.9 ± 129.7 ^a^	617.5 ± 267.0^a^	744.2 ± 144.0	698.4 ± 142.5	743.9 ± 162.3	**0.003**
Wanting (12–1200)	454.1 ± 244.2	366.2 ± 236.4	415.2 ± 242.8	303.6 ± 159.2	410.0 ± 235.8	0.114

HRV: heart rate variability; RMSSD: root mean square of successive differences; ANOVA for continuous normally distributed variables (mean ± SD), Kruskal–Wallis for continuous non-normally distributed variables (median (p25; p75)), chi-square for categorical variables (*n*; %). Groups with identical superscript letters are significantly different from each other.

**Table 2 nutrients-13-03654-t002:** Group differences in stress response and food parameters after Trier Social Stress Test induction.

	Normal Weight and Low Stress *n* = 52 (Mean ± SE)	Overweight and Low Stress *n* = 16 (Mean ± SE)	Normal Weight and High Stress *n* = 46 (Mean ± SE)	Overweight and High Stress *n* = 23 (Mean ± SE)
**Stress**				
% Stress reactivity	557.8 ± 200.8	564.9 ± 394.6	378.5 ± 214.7	725.6 ± 332.2
% Stress recovery	−32.1 ± 25.8	−39.3 ± 50.6	25.3 ± 27.5	26.6 ± 42.6
AUCi stress	−37.4 ± 25.6	−133.6 ± 50.3	−76.5 ± 27.4	−77.6 ± 42.4
% Negative emotions reactivity	19.9 ± 25.7	36.2 ± 50.4	54.6 ± 27.4	76.2 ± 42.4
% Negative emotions recovery	−0.9 ± 13.8	−15.3 ± 27.1	1.0 ± 14.7	21.2 ± 22.8
AUCi negative emotions	−33.5 ± 19.2	−17.4 ± 37.7	−37.5 ± 20.5	−61.1 ± 31.7
% Happy reactivity	160.6 ± 130.3	0.2 ± 256.0	50.5 ± 139.3	332.4 ± 215.6
% Happy recovery	**409.8 ± 148.4 ^a^**	75.0 ± 95.9	−8.8 ± 176.2	**−26.6 ± 89.7 ^a^**
AUCi happy	22.8 ± 27.6	13.6 ± 54.2	−2.1 ± 29.5	−22.2 ± 46.6
% Stress index reactivity	0.2 ± 4.2	3.2 ± 8.2	3.6 ± 4.6	5.6 ± 7.6
% Stress index recovery	2.2 ± 2.6	−0.9 ± 5.0	3.7 ± 2.8	4.4 ± 4.6
AUCi Stress index	−2.06 ± 3.3	−0.29 ± 6.3	−2.6 ± 3.6	4.1 ± 5.8
% RMSSD reactivity	−2.6 ± 6.5	−11.6 ± 12	−0.1 ± 7.1	−10.8 ± 11.6
% RMSSD recovery	3.4 ± 3.4	8.5 ± 6.6	7.0 ± 3.7	4.6 ± 6.1
AUCi RMSSD	−66.5 ± 36.8	−74.7 ± 71.1	−09.1 ± 39.9	−110.8 ± 65.7
% Salivary cortisol reactivity	**410.9 ± 1561.5 ^a^**	250.4 ± 3249.3	**137.7 ± 1636.8 ^b^**	**7856.9 ± 2710.9 ^a,b^**
% Salivary cortisol recovery	−49.4 ± 21.5	−6.9 ± 40.2	−27.1 ± 22.5	28.9 ± 34.0
AUCi Salivary cortisol	−19,806.6 ± 10,924.7	−9610.8 ± 22,170.7	−16,225.8 ± 11,752.7	−6059.5 ± 18,402.6
% Salivary alpha-amylase reactivity	4.8 ± 2.2	−2.4 ± 4.3	5.2 ± 2.3	5.9 ± 3.6
% Salivary alpha-amylase recovery	−4.7 ± 2.7	−2.0 ± 5.3	−3.3 ± 2.9	−4.1 ± 4.5
AUCi salivary alpha-amylase	−11,443.5 ± 1698.7	−8397.5 ± 33,667.4	−13,998.6 ± 18,324.7	27,116.6 ± 28,351.8
**Food Parameters**				
Increase in hunger (1–100)	13.0 ± 2.3	6.7 ± 4.7	14.0 ± 3.8	12.3 ± 3.8
Increase in wanting (12–1200)	89.0 ± 19.8	85.2 ± 39.6	93.0 ± 20.9	147.0 ± 32.4
Trait emotional eating (13–65)	26.3 ± 1.7	26.4 ± 3.4	30.2 ± 1.8	27.8 ± 2.8
Food intake (kcal)	347.6 ± 29.1	383.5 ± 58.2	331.5 ± 30.7	393.9 ± 47.7
Energy density intake (kcal/g)	2.5 ± 0.0	2.5 ± 0.1	2.4 ± 0.0	2.6 ± 0.1
HFSW intake (kcal)	**116.3 ± 13.6 ^a^**	170.6 ± 27.3	**134.7 ± 22.4^b^**	**189.6 ± 22.4 ^a,b^**
LFSW intake (kcal)	78.2 ± 7.7	87.9 ± 15.4	77.4 ± 8.1	83.0 ± 12.6
HFSA intake (kcal)	125.4 ± 14.7	116.2 ± 32.1	99.5 ± 16.3	99.7 ± 24.2
LFSA intake (kcal)	30.5 ± 3.8	22.3 ± 7.9	30.1 ± 4.0	20.6 ± 6.0

Linear regression estimated marginal means and standard errors (mean ± SE) adjusted for age, sex, and parental education. Wanting and snack buffet intake analyses were additionally adjusted for hunger at study start and liking of presented snacks. Groups with identical superscript letters are significantly different from each other (*p* ≤ 0.05). *p*-Values for % salivary cortisol reactivity, a = 0.019 and b = 0.016; *p*-values for % happy recovery, a = 0.014; *p*-values for HFSW intake (kcal), a = 0.007 and b = 0.041. AUCi: area under the curve with respect to increase; RMSSD: root mean square of successive differences; HF: high fat; LF: low fat; SW: sweet; SA: savoury.

## Data Availability

The datasets and codes generated during and/or analysed during the current study are available from the corresponding author.

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
