# Peer review of "Stress Responsiveness and Emotional Eating Depend on Youngsters’ Chronic Stress Level and Overweight"

_nutrients, 2021, doi:10.3390/nu13103654_

Round 1

Reviewer 1 Report

  1. Abstract & Introduction. The Authors announce to analyse „interrelated physiological mechanisms”, but then mention „stress responsiveness and eating behaviour” which are rather psychological mechanisms. Please make it sound less contradictory.
  2. „Stress is a threatened emotional condition” – not always, please consider distress vs eustress.
  3. The Authors excluded underweight children based on z-score. But z-score is sample dependent, and the sample is likely overweight as it was recruited from „the Obesity Prevention”. Saying that, the Authors likely excluded children with low to normal weight, but not underweight. Consequently, please do not exclude them using this rationale.
  4. Participants. Please also add a link to a Table displaying all sample characteristics (M, Mdn, Min, Max, SD)
  5. „Participant population”. The Authors analysed sample, not population.
  6. The Authors made categorisations for continuous variables (e.g., BMI, stress) what I consider unsupported. Continuous variables should be analyses using its full variability.
  7. The Authors treated sex a confounder, but it should be rather a factor. In adolescent age BMI is highly affected by physiologically changing body proportions (increase of muscles in boys; increase in fat in girls).
  8. Please also consider this work: Galasso, Letizia, et al. "Binge eating disorder: What is the role of physical activity associated with dietary and psychological treatment?." Nutrients12.12 (2020): 3622.

Reviewer 2 Report

The paper by Wijnant et al aims to investigate the role of chronic stress level and overweight on stress responsiveness demonstrating that the latter stimulated unhealthy and emotional eating. They evaluated salivary cortisol and alpha-amylase, linking the strong cortisol reactivity to higher fat/sweet snack intake.

The paper is quite interesting; however, it is very complex. It contains a lot of data and a lot of information and in the present form is difficult to read. I think that the paper needs to be simplified.

The introduction is too long, I suggest rewriting it in a more concise form.

The material and methods section is really too long. For example, in section 2.4.1 (page 5, lane 195-201), the first paragraph is not necessary. The same in the section 2.4.2 (page 5, lane 216-222). In the section 2.6.2 (page 7, lane 318-320), there is a sentence on spearman correlation on hair and salivary cortisol that seems to be a result. For the material and methods section, also, I suggest rewriting in a more concise form.

This suggestion (a concise form) is also for results and discussion.

In the results, paragraph 3.1 (page 8, lane 380-386) it is not indicated the table or figure in which are reported the results.

How the authors explain the low salivary cortisol in the OWHS group?

Minor comments:

There are some typing errors:

Page 16 lane 555, there is a parenthesis that is not necessary.

Page 17 lane 601, there is a reference number in superscript.

Page 17 lane 618, there is a full stop not necessary.

Round 2

Reviewer 1 Report

improved

Reviewer 2 Report

the authors replied to the comments